# Change in Convection Mixing Properties with Salinity and Temperature: CO_2_ Storage Application

**DOI:** 10.3390/polym12092084

**Published:** 2020-09-14

**Authors:** Lanlan Jiang, Sijia Wang, Donglei Liu, Weixin Zhang, Guohuan Lu, Yu Liu, Jiafei Zhao

**Affiliations:** Key Laboratory of Ocean Energy Utilization and Energy Conservation of Ministry of Education, Dalian University of Technology, Dalian 116024, China; lanlan@dlut.edu.cn (L.J.); wangsijia007@mail.dlut.edu.cn (S.W.); 13130409600@163.com (D.L.); 13130456518@163.com (W.Z.); WSJ875813767@163.com (G.L.)

**Keywords:** convection mixing, finger, dissolved mass, CO_2_ storage

## Abstract

In this study, we visualised CO_2_-brine, density-driven convection in a Hele-Shaw cell. Several experiments were conducted to analyse the effects of the salinity and temperature. The salinity and temperature of fluids were selected according to the storage site. By using charge coupled device (CCD) technology, convection finger formation and development were obtained through direct imaging and processing. The process can be divided into three stages: diffusion-dominated, convection-dominated and shutdown stages. Fingers were formed along the boundary at the onset time, reflecting the startup of convection mixing. Fingers formed, moved and aggregated with adjacent fingers during the convection-dominated stage. The relative migration of brine-saturated CO_2_ and brine enhanced the mass transfer. The effects of salinity and temperature on finger formation, number, and migration were analysed. Increasing the salinity accelerated finger formation but suppressed finger movement, and the onset time was inversely related to the salinity. However, the effect of temperature on convection is complex. The dissolved CO_2_ mass was investigated by calculating the CO_2_ mass fraction in brine during convection mixing. The results show that convection mixing greatly enhanced mass transfer. The study has implications for predicting the CO_2_ dissolution trapping time and accumulation for the geological storage of CO_2_.

## 1. Introduction

The excessive release of carbon dioxide (CO_2_) from industries has been a major cause of the rise of average surface temperature on the earth. As a substitute for conventional fossil fuel, renewable energy is a potential solution to reduce the emission of CO_2_. Renewable energy, especially solar and wind energy, is limited by intermittent nature environment [1,2,3]. Carbon capture and sequestration (CCS) is another alternative to keep CO_2_ concentration balance on earth. CCS is widely accepted as an optimal technique to prevent CO_2_ from entering the atmosphere [4]. Underground sequestration in geological formations is an option for the disposal of CO_2_ [5,6,7]. At present, potential carbon capture and sequestration (CCS) technologies include curing CO_2_ into inorganic carbonate, ocean storage and geological storage [8,9,10,11]. During the storage of CO_2_ in saline aquifers, the density of brine-saturated CO_2_ increases and results in a density difference between brine- and brine-saturated CO_2_. Therefore, density-driven convection mixing, caused by a difference in the density, enhances the dissolution velocity and spatial distribution of CO_2_, accelerates the transformation of the CO_2_ phases, reduces the possibility of CO_2_ leakage through capping layers, cracks, and abandoned wells, increases the effective storage capacity of CO_2_ in saline aquifers and increases the safety of long-term storage [12,13,14,15,16,17]. Understanding the convection mixing properties is extremely important for the long-term fate of the injected CO_2_ and the security of storage [18,19,20].

Several studies have analysed the startup and finger characteristics between two-phase interfaces during the convection mixing process using theory and simulations. The moment of the convection mixing startup is defined as the onset time, which reflects that the perturbation growth rate first becomes positive or a convection finger occurs [21,22]. Eniis-King et al. [23] estimated the onset time at the end of the diffusive mixing period in anisotropic systems by using linear stability analysis. CO_2_ diffused into brine and formed a diffusive boundary layer that gradually grew. Then, disturbances and fingers formed along the layers until the layer became sufficiently thick, reflecting the convection occurrence. Riaz et al. [24] performed numerical simulations with a linear stability analysis to predict the onset time and wavelength. The relationship between the onset time and convection instability properties, such as initial wavelength and growth rates, were obtained and demonstrated good agreement with high-accuracy direct numerical simulations. Pau et al. [25] carried out high mesh precision numerical simulations and compared their results with previous research results. The onset time was consistent with a previous linear stability analysis. These researchers also found that the CO_2_ flux at the top of the model reaches a stable state in time and space along with the beginning of convection mixing and was proportional to the permeability. The effect of background basic flow on the onset of convection was also numerically investigated, showing a proportional relation to the Rayleigh number (Ra) and transverse dispersion. However, most experiments cannot satisfactorily control the onset time because convection fingers were initially not detected [24] or do not show the physical properties of convection mixing and accelerated CO_2_ dissolution [26,27,28].

To visualise the convection mixing process from the onset time to development, Hele-Shaw cells [29] or 2D-etched pore networks [30] were used in lab-scale experiments. Hele-Shaw cells are thin cells between two parallel flat plates used to investigate natural convection of two different fluids. The advantages of Hele-Shaw cells are that their transparent appearance is convenient for a camera to capture images and their simplified 2D structure is available to reinforce to simulate CO_2_ convective flows under high temperature and high pressure condition [29]. The development of convection mixing depends on the fluid and rock physical properties, such as the density, viscosity, porosity and permeability, which are influenced by the salinity, temperature and pressure [31]. Additionally, the size and location of the convection finger and the growth were impacted by the flow field heterogeneity, the flow induced by fingering and the large-scale flow [32]. With the help of an optical technique, such as the Schlieren technique [33], convection dynamics were also visualised covering both buoyancy and surface-driven flows [34,35,36]. The convection mass with different concentrations of fluid pairs was examined to analyse its influence, showing that the convection mass has scaling properties of Ra^4/5^. The nonlinear relationship of the convection flux was explained by the proportional parameters of the convection flux and radial diffusion in the descending plume. Among them, most experimental studies have focused on the interface between either fully miscible or immiscible solvents with low Ra values using a Hele-Shaw cell during the early flow stage and have rarely investigated the entire dissolution process [37,38]. The results of laboratory experiments in deep layers should be applicable to the geophysical phenomena at high Ra values, and the natural onset of convection should also occur at the millimetre or centimetre scales [39]. Water and propylene glycol were used to replace CO_2_ and brine, respectively, to produce an approximately small density difference. The results show that the development and the characteristics of the convection dynamics are not the same quantitatively. The characteristics are different from immiscible and miscible systems typical of at CO_2_ water interface for which the evolution of the density stratification in the host phase is solubilisation-limited and not diffusion-limited.

In our previous study, we measured the convection properties with a denser and lighter fluid instead of CO_2_ and brine by using a Hele-Shaw cell and magnetic resonance imaging (MRI) technology [40,41]. In this study, we visualised the density-driven convection of CO_2_-brine in a Hele-Shaw cell at 0.1 MPa and various temperatures. The results have implications for predicting the dissolution trapping time and accumulation in the geological storage of CO_2_.

## 2. Materials and Methods

### 2.1. Experimental Apparatus and Materials

A schematic diagram of the experimental setup is shown in Figure 1, including a Hele-Shaw cell, a charge-coupled device (CCD) image acquisition system, a light emitting diode (LED, BXYZ, Beijing, China) backlight and a data acquisition computer connected to the CCD. The CCD camera model is a Microvision MV-E800M (BXYZ, Beijing, China), which can capture images with a 3312 × 2496 pixel resolution. The light source board was 25 × 25 cm^2^ in length and width. To keep the experimental system on a horizontal line, a loading platform was setup to adjust the height of the Hele-Shaw cell. The whole experiment was placed in a 140 × 60 × 70 cm^3^ rectangular insulated black box.

The Hele-Shaw cell consisted of five pieces of organic glass. The inner dimensions of the cell were 140 mm in width by 200 mm in height. The Hele-Shaw cell in Figure 2 was used with two parallel plates of organic glass. The gap b was controlled by thin stainless-steel shims, chosen here to be 1 mm. The Hele-Shaw configuration provides a useful model following the Darcy description with an isotropic permeability K = b^2^/12 [42] and a porosity of unity (φ = 1).

The viscosity was measured by a traditional viscometer (NDJ-8S, Shanghai Pingxuan Scientific Instrument Co., LTD, Shanghai, China). Experimental temperatures were chosen as 33 °C, 38 °C and 43 °C, corresponding to the various salinities of brine in underground conditions. All of the experiments were conducted at 0.1 MPa. All setups were placed in a dark box to prevent disturbances. A closed thermal insulation box was used to control the system temperature by using heating plates. A temperature controller (AT72AAS4R, Autolise Co., LTD Dalian, China, 0.1 °C) increased the temperature with a large power supply.

Salinity has been observed to increase with depth in many sedimentary basins or at least to display stratification into the subhorizontal layers. Salinity measurements of water samples indicate density differences ranging from 0.1 to 5.1 kg/m^3^ in the top 100 m of the Yarragadee Aquifer [43], with most values being less than 3 kg/m^3^. The lowest values occur in areas of groundwater recharge in the aquifer. In this study, NaCl solutions with various salinities of 0, 0.25, 0.50, 1.00 and 2.00 wt% were chosen, with density differences less than 1.5 kg/m^3^. To distinguish the NaCl solution, the coloured dye bromocresol green was added before the experiment, which changed with the pH value. For our experimental equipment, the optimum concentration of bromocresol green was 2.5 × 10^−4^ mol/L, which has no effect on the properties of the water. The fluid parameters, such as the density and viscosity, are listed in Table 1.

### 2.2. Experimental Procedures

Before CO_2_ injection, the temperature was kept constant in the gas pump by the water bath. The focal length of the CCD camera and the brightness of the LED light source were optimally adjusted to acquire high-resolution images. Then, CO_2_ was injected from the top side of the Hele-Shaw cell. The air in the top side of the cell was displaced out by the CO_2_. The CO_2_ in contact with the brine solution gradually dissolved into the brine. The pH value of pure water changed from 5.6 to 3.9 under full CO_2_ saturation conditions. The pH variation during CO_2_ injection can be indicated with the acid and base indicator of bromocresol green. The CCD continued to acquire images with a time interval of 10 s.

### 2.3. Dimensionless Parameters

To better understand the density-driven convection mechanism, some dimensionless parameters were used to describe and compare the experimental results for different scales and conditions. As mentioned above, convection mixing due to gravity instabilities has been characterised in detail elsewhere [44] using the *Ra*, which is the ratio between the buoyancy and diffusion force and is given by
(1)$$Ra=\frac{k\Delta \rho gH}{\mu D\phi }$$
where *k* is the permeability of the porous medium; Δρ refers to the maximum density difference between the fluids; *H* is the characteristic length of the system, i.e., the vertical length of the domain; *μ* is the viscosity of the light fluid; *D* is the diffusion coefficient and ϕ is the porosity of the porous medium. In this study, the Ra ranged from 6.536 × 10^4^ to 6.940 × 10^4^.

## 3. Results and Discussion

### 3.1. Convection Mixing Occurrence and Development

The convection mixing caused by a density difference in pure water (0 wt%) at 33 °C is shown in Figure 3. The boundary between CO_2_ and the brine solution is clearly shown in the images. With time lapses, the brightness of brine changes from dark to white due to CO_2_ dissolution. This change means that three-phase fluids, such as CO_2_, brine and brine saturated with CO_2_, can be distinguished clearly from the CCD technology. The formation and development of fingers can be found throughout the images. With CO_2_ dissolving into brine, grey fingers are distributed in the solution phase. Similar to previous studies [40,41], three stages were found during the convection mixing process from the images: Stage I (diffusion-dominated stage,Figure 3a,b), Stage II (convection-dominated stage, Figure 3c,d) and Stage III (shutdown stage, Figure 3e–h). The corresponding colour bar is added to the right of this figure, where 0 represents a pure salt solution and 1 represents a salt solution that completely dissolves carbon dioxide. And the 0 and 1 on the colour bars of the other figures have the same meaning as in Figure 3.

#### 3.1.1. Diffusion-Dominated Stage and Convection Finger Formation

As CO_2_ is injected, it diffuses gradually into brine with a slight colour change along the interface. Even in real geological reservoirs, injected CO_2_ gradually diffuses into brine, which takes up several years. A local equilibrium is reached along the boundary, reflecting the thin diffusion layers. A thin diffusion layer forms at the boundary between two fluids during the early stage, as shown in Figure 3a. CO_2_ is concentrated along the layer in the solution. We define this stage as diffusion-dominated since gaseous CO_2_ migration is dominated by self-diffusion. Because of the low diffusion coefficient up to 9 orders of magnitude, the diffusion layer becomes thicker and diffuses at a slow rate. Finally, late diffusion and the transition occur when the diffusion front grows to a certain thickness and the initial fingers grow independently.

In this study, the occurrence of fingers reflects the start of convection mixing. The moment of the startup represents the onset time of convection mixing. Generally, the onset time is a function of the permeability, fluid properties and other parameters. The moment reflects the mass transfer change from diffusion to convection mixing. For pure water, the diffusion layer is demonstrated up to 50 s, while the finger is exhibited at 130 s. For CO_2_ injection, the onset time is related to the safety of the geological storage. The shorter the onset time is, the greater the storage safety is.

#### 3.1.2. Convection Finger Development to the Shutdown

After the finger grew along the boundary, as shown in Figure 3b, the convection developed into the convection-dominated stage, namely, stage II. In the near boundary zone, each finger grows longer and progresses deeper into the brine phase. Meanwhile, small fingers continuously form along the boundary. Therefore, in stage II, the formed fingers grow rapidly and then aggregate with adjacent fingers. The aggregation fingers move down to the bottom of the cell. However, this is not the end of convection finger development. After the fingers strike the bottom, they develop horizontally and mix fully within the whole cell.

During the rapid growth process, small fingers form along the boundary in a short time. The fingers form along the boundary and are distributed differently with salinity and temperature. For example, at 360 s and 33 °C (Figure 3c), the finger is distributed uniformly along the boundary, reflecting the limitation of the finger disturbance. The finger number (FN) gradually increases until all fingers are distributed along the interface.

The fingers expand and progress downward due to the density-difference-driven force. In theory, the mass transfer of CO_2_ and brine transforms from diffusion-dominated into convection-dominated. As time progresses, adjacent fingers aggregate into large fingers and simultaneously displace the brine in the cell. Interestingly, the finger shapes change from a small drop into a streamline. The front of the finger is rich and bulky, while the tail of the finger is thin and small. The variations in the shape follow the convection development. During the convection-dominated stage, it can be found that CO_2_ downward migration is coupled with brine. At 2000 s, as shown in Figure 3d, brine migration is clearly marked by white arrows. The migration area is wide at 4000 s and then shrinks into a small area at 8000 s. Brine migration remains in a single and narrow path for a long time. The relative migration influences both finger development and brine migration. Brine migration can be blocked by finger falling. From the figure, the relative migration shows interesting bending phenomena, and a single finger divides into multiple fingers, as shown in Figure 3g,h. These phenomena inhibit the mass transfer of CO_2_ and brine further. Finally, the process progresses into the shutdown stage without any convection mixing. The duration time for each stage is different for all cases.

Notably, tiny bubbles form along the wall. CO_2_ first dissolves into the brine layer and then forms a precipitin below the layer, as shown in Figure 3g,h. The precipitin bubble concentrates into the wall and increases because of convection. Precipitin formation is useful for CO_2_ storage and can increase the safety of the geological storage. In the injection well, CO_2_ dissolves in a short time, then precipitates and is trapped in the small pore spaces at far injection well.

### 3.2. Effect of Brine Salinity on the Onset Time

Figure 4 shows finger formation and development during convection mixing for all cases with different salinities and temperatures. To clearly distinguish the finger in the brine phase, the grey images were rendered with pseudocolour using ImageJ software (NIH, Bethesda, MD, USA). Brilliant yellow was added to the fingers. The onset time varies with salinity, as shown in Figure 4. At the same temperature, such as 33 °C, the onset time was 130 s for pure water (0 wt%), 100 s for 0.25 wt% saline and 20 s for 1.00 wt% saline. As the salinity increased, the brine density increased within 2.71%, and the maximum density difference between CO_2_ and brine changed from 996.4 to 1024.3 kg/m^3^, while the viscosity decreased within 1.92%. Therefore, the onset time became shorter with increasing salinity. These onset time results agree well with the previous study, which reported an inverse function with the density difference and salinity and a positive function with the viscosity and diffusion coefficient [40,41].

To generalise the analysis of the influencing factors, Figure 5 shows that the onset time inversely changed with the Ra. The overall trends for other cases with various temperatures are similar, showing that the onset time increases with decreasing Ra. The influence of the Ra on the onset time can be divided into low- and high-salinity regions. With low salinity, the onset time is obviously higher than in other regions. The onset time decreases, and the finger develops into a single finger. However, convection mixing never occurs if salinity increases too much.

Compared to a high-density fluid, the onset time of a low-density fluid is 6–10 orders of magnitude larger. If the salinity is larger than 1.00 wt%, then the boundary is extremely unstable, and convection mixing occurs almost instantaneously with CO_2_ injection. Hence, the onset time can be as short as 20 s. It can be concluded that increasing the salinity effectively changes the onset time of convection in a nondesirable way. The salinity has a negative effect on convection mixing with an increase in the salinity, variables such as the density difference and diffusion coefficient will decrease, and the viscosity of the fluid will increase, but their total effects facilitate a reduction in the Ra. The increasing salinity results in an increase in the viscosity of water since the dissolved ionic compounds increase the density of water and its viscosity. Ionic compounds accomplish this feat by organising water molecules around the ions using ion-dipole interactions; that is, water is the dipole, and the dissolved ions are the ions.

### 3.3. Effect of Brine Salinity on Finger Development and Mixing

A dimensional parameter, finger number (FN), was determined to analyse the effect of the salinity on convection mixing. The number was counted by image processing with ImageJ software [40,41]. Figure 6a shows the variation curve for the number of fingers at the initial interface at 33 °C.

The results show that the FN increases sharply to a peak value in a short period of time (within 600 s), then is maintained during the finger development for a long period, and finally decreases gradually due to margination. For all the cases with different salinities, the change in the FN showed a similar trend. The peak of the FN decreases with increasing salinity, while a slight difference occurred within pure water and 0.25 wt%, and a larger difference occurred in 0.5 wt%. The difference is caused by the onset and falling of the fingers and agrees well with their development stage. Adjacent fingers touch and merge, leading to a decrease in the FN. The more rapidly the fingers merge, the steeper the FN decrease is with an increase in the salinity. However, there are fluctuations as the FN decreases because of the appearance of new fingers and the splitting of fingers. During convection mixing, more FNs mean a larger interface area between the fluid phases in the cell, which is better for the mass transfer of CO_2_ and brine during storage.

Figure 6b compares the maximum finger number with Teng’s and Jiang’s data, showing our data are smaller (about 37.17%) than Teng’s and Jiang’s data [45,46]. The reason may be that both of them adopt simulation fluids for CCS and thermodynamics property of which differ from the real CO_2_ gas we used.

As seen in Figure 7, an increase in salinity suppresses finger movement. For example, at 4000 s, a convection finger moves downward to the center of the vessel at a 0.25 wt% salinity, while the finger reaches the bottom at 0 wt%. At 1 wt%, the finger just begins expanding into the vessel. Therefore, both the development rate and movement rate of fingers are dominated by the salinity. The density and density difference increase with salinity. As the convection finger forms and moves downward, the fresh brine on the bottom side moves upward. However, the slow movement rates limit further contact at the boundary. The phenomenon causes the finger to develop into a large shape with a small FN density. With the salinity increases to 2.00 wt%, the layer becomes thicker but without any disturbance, which means convection mixing never occurs at a high salinity. The results directly demonstrate that the salinity inhibits the formation and development of convection mixing and is then averse to the mass transfer of CO_2_.

As mentioned above, convection mixing occurs only when salinity does not exceed the threshold, and in this experiment that is 2.00 wt%. Salinity can suppress fingers’ occurrence and propagation, which are the main appearance of convection mixing. Increasing salinity also decreases the maximum finger number, in other words the mass transfer area declining. Thus, it is of great importance to choose a suitable salinity range in CO_2_ storage application.

### 3.4. Effect of Temperature on the Onset Time and Convection Finger

To investigate the effect of temperature, the same experiments were carried out at 38 C and 43 °C Binary images for finger development are shown in Figure 8. Figure 9 shows the percentage change in sweep area of CO_2_ solution obtained from three repeated experiments at 33 °C with zero salinity. This percentage approximately reflects the ratio of CO_2_ solubility to the CO_2_ saturation in the Hele-Shaw box if the inhomogeneity of CO_2_ distribution in the finger is ignored. The approximate saturation of CO_2_ is close to 100% after 4 h. At low salinity, the FN decreases with temperature and then increases. However, the shape of the finger becomes thinner. This effect may be because the viscosity of the liquid decreases. The finger area in the brine phase decreases, and even the FN increases with temperature. The influence of temperature seems to be complex. The CO_2_ solubility decreases with increasing temperature. The onset time decreases with temperature, while the grey value of the images shows an inverse rule. This result means that the temperature inhibits the mass transfer of CO_2_. At 0.25 wt% saline, the FN increases with temperature. The temperature sensitivity is larger than that at a low salinity. With the same temperature difference, the density difference is different at various salinities. However, the temperature suppresses the FN as the salinity increases to 1.00 wt%. Even the convection mixing completely disappeared at high temperature and a 1.00 wt% salinity, as shown in Figure 10.

### 3.5. Dissolved Mass CO_2_

The most practically significant measure of the dissolution process is the dissolved flux, which quantifies the rate of the change in the total CO_2_ dissolution in brine. The dissolution flux per unit area can be obtained from the dissolved mass. To measure the dissolved mass of CO_2_, the mass fraction of CO_2_ was used for the quantitative analysis by calculating the ratio of the dissolved CO_2_ area into the cell area of brine. By calculating the grey value ratio of brine saturated CO_2_ to the total intensity, the fraction was obtained to reflect the dissolved mass of CO_2_.

The mass fraction at 33°C is shown in Figure 11. The mass fraction of dissolved CO_2_ was calculated with repeated experiments. In pure water (0 wt% saline), the fraction linearly increased with time until 8000 s, and the slope change was caused by a finger falling down to the bottom of the cell. The movement weakens the vertical finger speed, which that the turning point at 8000 s is the boundary from the development region to the stable region of the convection process.

The diffusion coefficient of CO_2_ into water is evaluated by the theory of pressure decay and is used as a key parameter to quantify convection and its effect on the mass transfer of CO_2_ [47]. Compared to the mass transfer dominated by diffusion, the dissolved mass of CO_2_ at 18,000 s is far from 0.1% [47]; however, as the temperature increases, the mass of CO_2_ decreases. In this study, the dissolved mass of CO_2_ reached 75% within 8000 s. The results indicate that convection mixing greatly enhanced the mass transfer. Similar results were found in previous studies [48], showing that the rate of mass transfer in the dissolution mechanism cannot be analytically modelled by using the diffusion coefficient because density-driven convection enhances the rate of gas dissolution.

The influence of salinity on the dissolved mass of CO_2_ was also analysed, as shown in Figure 10. The dissolved mass of CO_2_ strongly depends on the salinity. As the salinity increases, the fraction decreases substantially. For pure water, the mass fraction accounts for 50%; however, it shrinks to 25%. For a 1.00 wt% salinity, the fraction kept rising gradually and softened. The overall curve reveals that the mass fraction is enlarged within 1% and then shrinks; this is the critical salinity in this study. For a 2.00 wt% salinity, the mass fraction of dissolved CO_2_ gradually increases. Diffusion dominated the mass transfer, and the increase rate of the mass fraction exhibited the same behaviour as the diffusion coefficient. The rate of increase slowly decreases and finally reaches that of the diffusion coefficient. For a 1.00 wt%, the mass fraction of dissolved CO_2_ increases at a constant rate. The reason was the dynamic equilibrium between convection and diffusion. The results again demonstrated the critical concentration in the study. Due to salinity differences between fluids, the total mass transfer decreases as the salinity increases. During the early stage, the diffusion was similar in each case with different salinities. However, the diffusion coefficient differs with convection in the later stage, and the differences become large.

## 4. Conclusions

In this study, we visualised CO_2_-brine, density-driven convection mixing in a Hele-Shaw cell at various temperatures and salinities. Convection mixing formation and development were visualised by CCD technology, and impact factors such as temperature and salinity were analysed in detail.

Three stages were found during the convection mixing process from the images: Stage I (diffusion-dominated stage), Stage II (convection-dominated stage) and Stage III (shutdown stage). A thin diffusion layer forms at the boundary between two fluids during the early stage. After the finger grew along the boundary, the convection developed into a convection-dominated stage. The formed fingers grow rapidly and then aggregated with adjacent fingers. Meanwhile, the small fingers continuously formed along the boundary. The CO_2_ downward migration was upward-coupled with brine.

The onset time was 130 s for pure water, 100 s for a 0.25 wt% salinity, and 20 s for a 1.00 wt% salinity at 33°C. The onset time became shorter with increasing salinity. The salinity had a negative effect on convection mixing. Compared to a high-density fluid, the onset time of a low-density fluid is 6-10 orders of magnitude larger. However, if the salinity increases too much, convection mixing never occurs. A short onset time is useful for the safe storage of CO_2_ storage. The FN increases sharply into a peak value in a short period, then remains during finger development for a long time, and finally decreases gradually due to margination. The more rapidly fingers merge, the steeper the FN decrease is with increasing salinity. Finger development during convection mixing is also influenced by the salinity. The increase in salinity suppresses the movement of fingers. Both the development and movement rates of fingers are dominated by the salinity.

Moreover, the effect of temperature on convection mixing is complex. If the salinity is high, then the sensitivity of temperature is high. The dissolved mass of CO_2_ was investigated by calculating the mass fraction of CO_2_ in brine during convection mixing. Compared to the mass transfer dominated by diffusion, the dissolved mass of CO_2_ at 18,000 s is far from 0.1% and reaches 75% within 8000 s. The results mean that convection mixing greatly enhances the mass transfer. The dissolved mass of CO_2_ strongly depends on the salinity. The study has implications for predicting the CO_2_ dissolution trapping time and accumulation for the geological storage of carbon dioxide.

## Figures and Tables

**Figure 1 polymers-12-02084-f001:**
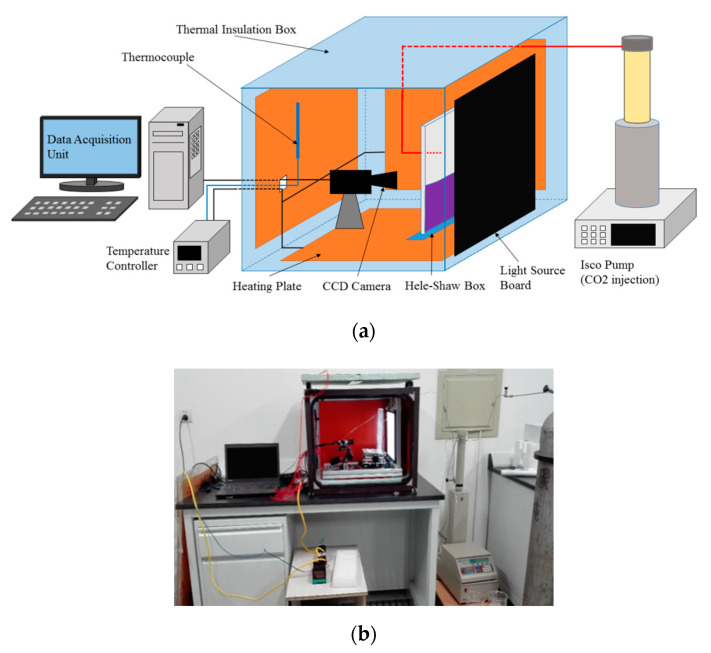
A system diagram of the experimental setup (**a**) and a photo (**b**).

**Figure 2 polymers-12-02084-f002:**
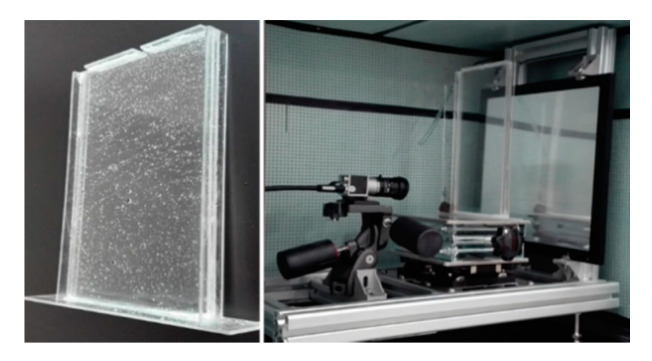
Photo of the Hele-Shaw cell.

**Figure 3 polymers-12-02084-f003:**
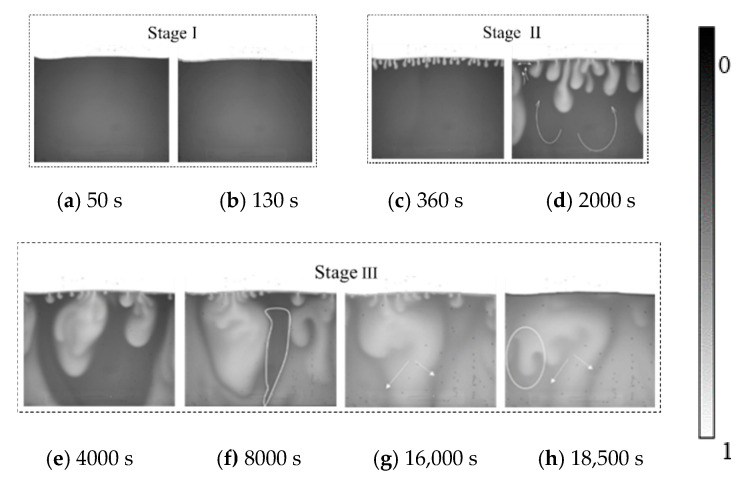
Convection fingers in pure water at 33 °C. (**a**) 50 s, (**b**) 130 s, (**c**) 360 s, (**d**) 2000 s, (**e**) 4000 s, (**f**) 8000 s, (**g**) 16,000 s and (**h**) 18,500 s.

**Figure 4 polymers-12-02084-f004:**
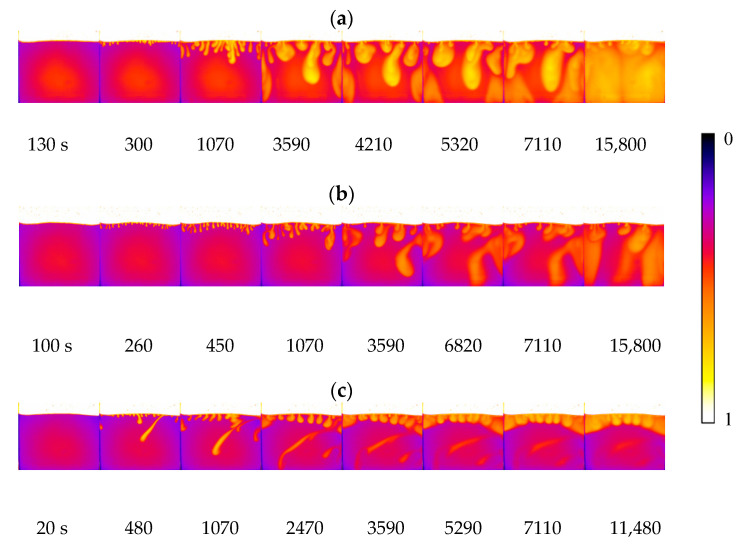
Convection development at 33 °C and various salinities: (**a**) 0 wt%, (**b**) 0.25 wt% and (**c**) 1.00 wt%.

**Figure 5 polymers-12-02084-f005:**
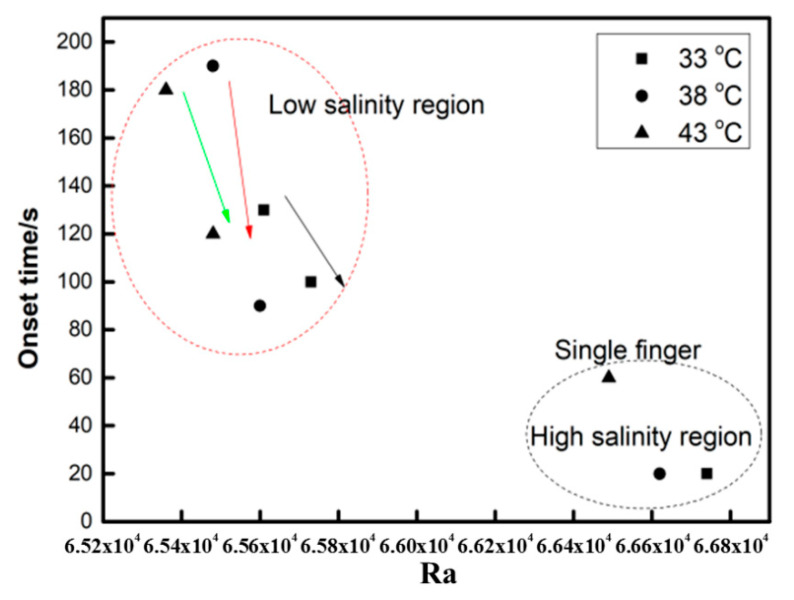
Variation in the onset time with the Ra.

**Figure 6 polymers-12-02084-f006:**
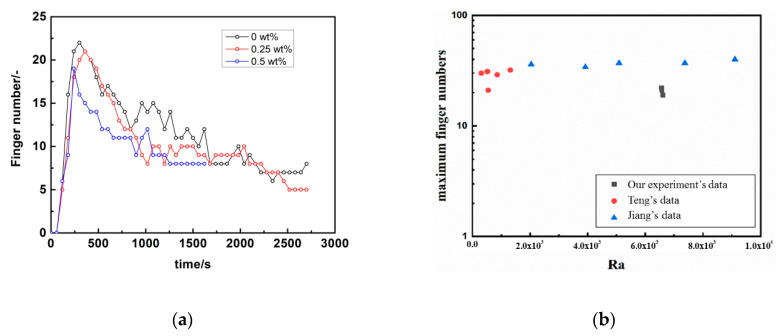
(**a**) Convection finger number changes over time at 33 °C; (**b**) comparison data of maximum finger number with others’ work.

**Figure 7 polymers-12-02084-f007:**
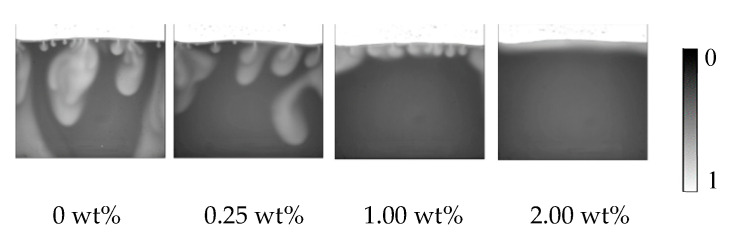
Difference in the fingers at different salinity at 4000 s and 33 °C.

**Figure 8 polymers-12-02084-f008:**
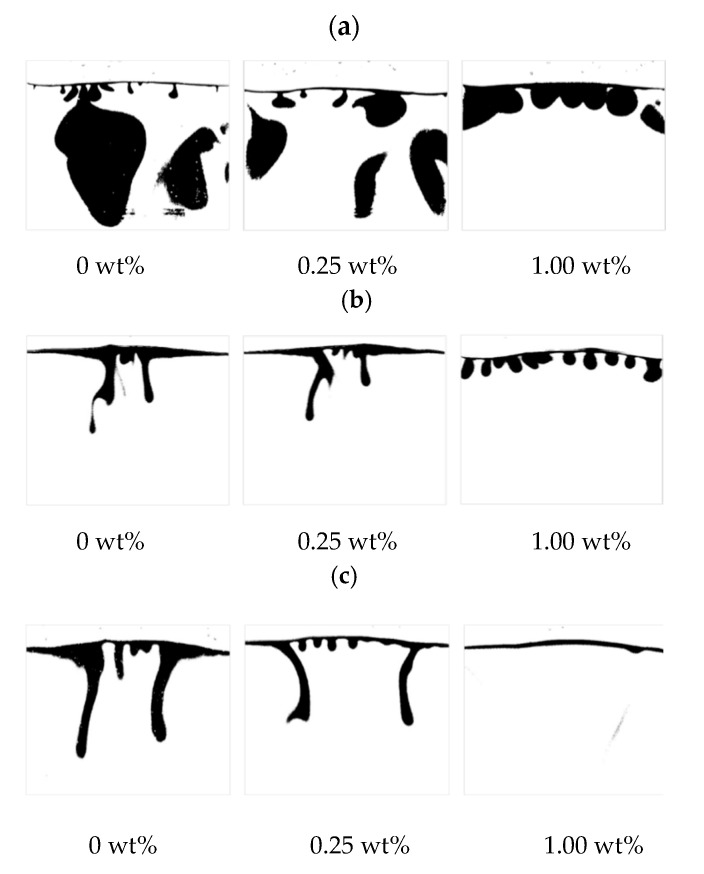
Convection finger changes with temperature at 10,000 s: (**a**) 33 °C, (**b**) 38 °C and (**c**) 43 °C.

**Figure 9 polymers-12-02084-f009:**
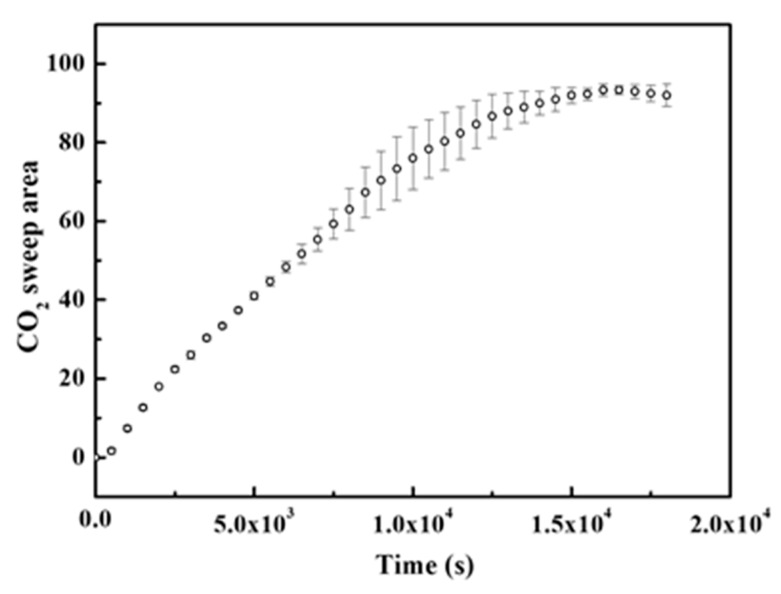
CO_2_ sweep area changes with time at 33 °C, atmospheric pressure, and 0 salinity.

**Figure 10 polymers-12-02084-f010:**
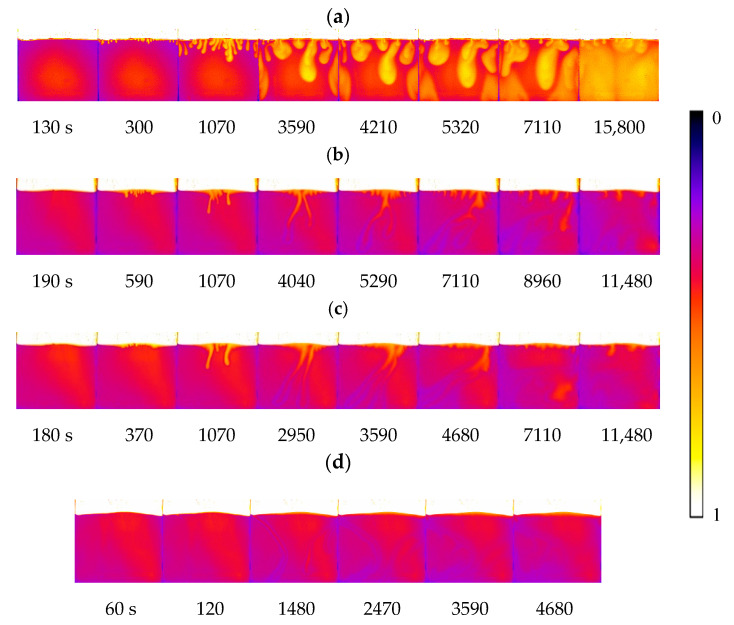
Convection development at a 0 wt% salinity: (**a**) 33 °C, (**b**) 38 °C and (**c**) 43 °C and (**d**) a 1.00 wt% salinity at 43 °C.

**Figure 11 polymers-12-02084-f011:**
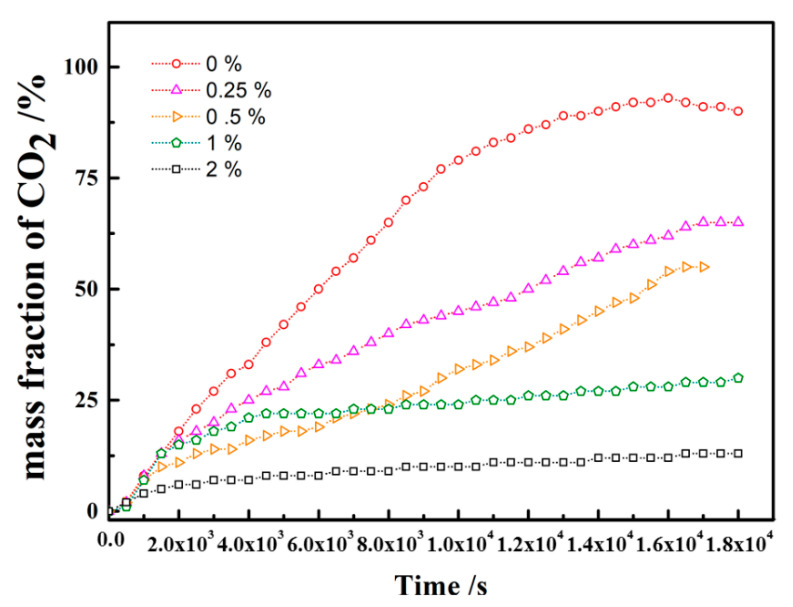
The mass fraction of CO_2_ during the convection change with salinity.

**Table 1 polymers-12-02084-t001:** Physical parameters of the fluids in the experiments.

Test No.	Temperature (K)	H (cm)	Salinity (%)	Δρ (kg/m^3^)	Viscosity (10^−3^ Pa·s)	Ra
1	306.15	20	0.00%	1.001	1.04	6.561 × 10^4^
2			0.25%	1.003	1.04	6.573 × 10^4^
3			0.50%	1.005	1.035	6.617 × 10^4^
4			1.00%	1.008	1.03	6.674 × 10^4^
5			2.00%	1.014	1.01	6.844 × 10^4^
6	311.15		0.00%	0.999	1.04	6.548 × 10^4^
7			0.25%	1.001	1.04	6.560 × 10^4^
8			0.50%	1.003	1.035	6.605 × 10^4^
9			1.00%	1.007	1.03	6.662 × 10^4^
10			2.00%	1.012	1.01	6.832 × 10^4^
11	316.15		0.00%	0.997	1.04	6.536 × 10^4^
12			0.25%	0.999	1.04	6.548 × 10^4^
13			0.50%	1.001	1.035	6.592 × 10^4^
14			1.00%	1.005	1.03	6.649 × 10^4^
15			2.00%	1.028	1.01	6.940 × 10^4^

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
