# Peer review of "Change in Convection Mixing Properties with Salinity and Temperature: CO2 Storage Application"

_polymers, 2020, doi:10.3390/polym12092084_

Round 1

Reviewer 1 Report

The manuscript sheds light on the CO2-brine convective mixing in a Hele-Shaw cell at different temperatures and salinities. The authors have visualized the development of the convective mixing and three stages are introduced to describe the process.

In my view, the paper makes a contribution in characterizing the convective mass transfer happing during the storage of CO2 in saline aquifers. The authors explain every detail of their approach and the main goal of their study is clearly stated.   

With appropriate consideration to the suggested comments, as well as other reviewers' comments, I find this paper appropriate for publication in the Polymers Journal.

  • Line 13: Please specify what the “CCD” stands for?
  • Line 129: “the temperature was kept in the gas pump” -- > “.. was kept constant ?”
  • What is the resolution of images captured by the camera? It should be mentioned in the manuscript.
  • Lines 159-160: I would suggest showing different stages in Fig. 3.
  • 4. A color bar should be added to the figure to show the concentration range. The same for Fig. 7 and Fig. 9.
  • Line 324: How the theory of pressure decay is used to calculate the diffusion coefficient. The description should be added to the manuscript.
  • In general, the authors should compare their results with past experimental and simulation studies in the literature.

Author Response

Thank you for your sincere advice, a point-by-point response letter have upload as a world file.

Reviewer 2 Report

Comments on “Change in convection mixing properties with salinity and temperature: CO2 storage application”

1-  English needs more work. It is not a high quality manuscript and there is a room for further improvement of writing.

2-  Authors should conduct a thorough literature review on advanced thermal engineering in solar collectors, storage systems and reflect the state of the art research in this area as well. Introduction needs to be enriched with more related studies. Searching the literature, following papers are suggested to be read and used:

-Numerical simulation of natural convection heat transfer of nanofluid with Cu, MWCNT, and Al2O3 nanoparticles in a cavity with different aspect ratios

-Thermodynamic potential of a high-concentration hybrid photovoltaic/thermal plant for co-production of steam and electricity

-Diurnal thermal evaluation of an evacuated tube solar collector (ETSC) charged with graphene nanoplatelets-methanol nano-suspension

4- Authors should represent a clear definition of Hele-Shaw in the manuscript, somewhere in the introduction.

5- Why CO2 storage is importance? Considering the limited space available on earth, why did not consider dissociation of CO2 or conversion of CO2 into value-added products?

6- A robust uncertainty analysis is required for the test rig used in the experiment.

7- Again, it was not clear how mixing affected the performance of the system. This needs further clarification in the paper.

Author Response

(The authors gave the same response as above.)

Round 2

Reviewer 2 Report

Everything is great. Just please check the format of references. "2" in "CO2" should be subscript.